# Neuroinflammation in Sepsis: Molecular Pathways of Microglia Activation

**DOI:** 10.3390/ph14050416

**Published:** 2021-05-01

**Authors:** Carolina Araújo Moraes, Camila Zaverucha-do-Valle, Renaud Fleurance, Tarek Sharshar, Fernando Augusto Bozza, Joana Costa d’Avila

**Affiliations:** 1Immunopharmacology Lab, Oswaldo Cruz Institute, Oswaldo Cruz Foundation, Rio de Janeiro 21045-900, Brazil; carolinamoraes0203@gmail.com; 2National Institute of Infectious Disease Evandro Chagas, Oswaldo Cruz Foundation, Ministry of Health, Rio de Janeiro 21040-360, Brazil; camila.valle@ini.fiocruz.br (C.Z.-d.-V.); bozza.fernando@gmail.com (F.A.B.); 3UCB Biopharma SRL, 1420 Braine L’Alleud, Belgium; Renaud.Fleurance@ucb.com; 4Experimental Neuropathology, Infection, and Epidemiology Department, Institut Pasteur, 75015 Paris, France; tsharshar@gmail.com; 5Université de Paris Sciences et Lettres, 75006 Paris Paris, France; 6Neuro-Anesthesiology and Intensive Care Medicine, Sainte-Anne Hospital, Paris-Descartes University, 75015 Paris, France; 7D’Or Institute for Research and Education, Rio de Janeiro 22281-100, Brazil; 8School of Medicine, Universidade Iguaçu, Rio de Janeiro 26260-045, Brazil

**Keywords:** brain, sepsis-associated encephalopathy, neuroinflammation, microglia, inflammasome, oxidative stress, neurotoxicity, synaptic dysfunction

## Abstract

Frequently underestimated, encephalopathy or delirium are common neurological manifestations associated with sepsis. Brain dysfunction occurs in up to 80% of cases and is directly associated with increased mortality and long-term neurocognitive consequences. Although the central nervous system (CNS) has been classically viewed as an immune-privileged system, neuroinflammation is emerging as a central mechanism of brain dysfunction in sepsis. Microglial cells are major players in this setting. Here, we aimed to discuss the current knowledge on how the brain is affected by peripheral immune activation in sepsis and the role of microglia in these processes. This review focused on the molecular pathways of microglial activity in sepsis, its regulatory mechanisms, and their interaction with other CNS cells, especially with neuronal cells and circuits.

## 1. Introduction

Sepsis is defined as a life-threatening organic dysfunction caused by a dysregulated host response to infection [1]. It was estimated that in 2017, there were 48.9 million cases and 11 million sepsis-related deaths worldwide, with a mortality rate that varies from 26% to 60% depending on the severity [2]. Despite all the efforts to improve sepsis treatment in the last decades, there is no specific therapy for sepsis, and mortality remains high, especially in low- and middle-income countries.

Sepsis is a clinical syndrome characterized by a maladaptive host response to infection that may be significantly amplified by endogenous factors. The pathogenesis of sepsis goes beyond an uncontrolled inflammatory response and includes major modifications in nonimmunologic systems such as cardiovascular, autonomic, neuroendocrine, coagulation, bioenergetic, and metabolic alterations [1]. While the inflammatory response in sepsis is relatively well-understood, the mechanisms driving metabolic deregulation and multiple organ dysfunction remain puzzling.

Multiple organ dysfunction is a hallmark of sepsis pathogenesis, and neurological manifestations are a frequent and underestimated symptom. Sepsis induces an acute brain dysfunction that is not related to direct brain infection and is characterized by various clinical and electroencephalographic changes [3,4]. This acute brain dysfunction is known as sepsis-associated encephalopathy (SAE), and it ranges from sickness behavior to altered consciousness, varying from confusion to delirium and, in worse cases, even coma [5]. SAE is strongly associated with higher mortality [6] and long-term cognitive impairments [7,8] affecting patients’ quality of life. To date, no curative or preventive treatment for SAE exists, and some pharmacological attempts have proved ineffective or even dangerous [9]. The absence of an effective therapeutic strategy is explained by the lack of precise knowledge of pathophysiological mechanisms. Some authors even suggest that the neurocognitive sequelae identified in the patients who had stayed in an ICU could be a risk factor for progressive neurodegenerative disorders of higher functions [10,11].

The mechanisms involved in SAE are diverse and include neurotransmitter dysfunction, inflammatory and ischemic lesions to the brain, microglia activation, changes in the blood–brain barrier (BBB) permeability, oxidative stress due to inflammation and mitochondrial dysfunction, and bioenergetic shifts that are part of metabolic adaptations to systemic inflammation [12,13]. Evidence suggests that alterations of the BBB resulting from both microglial and endothelial cell activation [14] could, in part, explain the acute neurological dysfunction by increasing the entry of inflammatory mediators and neurotoxic substances within the CNS [15]. Local production of pro-inflammatory markers and damage-associated molecular patterns (DAMPs) by the cells maintaining BBB (activated endothelial cells and astrocytes and perivascular macrophages) also contribute to neuroinflammation. Dysregulated cytokine responses are significant contributors to tissue injury and neurological deficits [16,17,18]. Inflammatory cytokines are involved in the pathophysiology of intensive care unit-acquired weakness, mostly, of neuropathies and myopathies [19]. All these mechanisms constitute a process known as neuroinflammation, which is the focus of this review.

To some extent, neuroinflammation is essential for maintaining brain homeostasis by inducing repair mechanisms. However, non-resolving neuroinflammation is a potentially harmful mechanism of brain damage. In this respect, microglia are major players in neuroinflammation, and exacerbated microglia activation has been associated with neurodegenerative diseases. As neurodegenerative diseases progress, microglia switch from a helpful role to a dysfunctional phenotype that becomes detrimental to neurons [20]. It is becoming clear that microglia surrounding Aβ plaques or Lewy bodies, for example, are not activated, but are nonfunctional [21]. There is a delicate balance determining whether microglia—and neuroinflammation in general—have beneficial or detrimental effects on the brain. This dual role is covered in the following sections, with a focus on sepsis and SAE.

## 2. Neuroinflammation in Sepsis

Neuroinflammation is defined as an inflammation within the CNS characterized by activation of neuroglial cells (microglia and astrocytes), increased inflammatory mediators in the cerebral parenchyma, and leukocyte recruitment, culminating in neuronal damage [22]. This intracerebral inflammation is a major driving factor for acute neurological outcomes in sepsis. Depending on the initial trigger, the neuroinflammatory response can evolve to resolution or chronification with multiple consequences.

The acute phase of sepsis is characterized by a massive systemic release of inflammatory mediators, DAMPs and PAMPs, that signal to the brain and induce neuroinflammation by multiple mechanisms (Figure 1). The inflammatory signals can reach the brain by different routes and in different regions by neural or humoral pathways that together orchestrate the inflammatory stress response clinically observable as the sickness behavior [16].

Inflammatory mediators are allowed to traffic from the blood into the brain through regions deprived of the BBB, reaching neuroendocrine and autonomic centers [13,16]. Brain regions such as the choroid plexus and circumventricular organs, especially the area postrema, are more permeable to inflammatory mediators that cross the BBB or signal through neurovascular units (NVUs), inducing microglia activation and local production of inflammatory mediators. The perivascular space in NVUs also houses innate immune cells involved in immune homeostasis and profoundly influences the brain’s response to infectious or immune challenges [23]. A recently described blood–brain communication mechanism shows that the release of extracellular vesicles by the choroid plexus epithelium cells during systemic inflammation induces pro-inflammatory miRNAs, including miR-146a and miR-155, in the CSF [24]. Neural pathways are rapid transmission routes through vagal afferents that innervate abdominal structures to the medullary autonomic nuclei, which connect to other autonomic neuroendocrine and behavioral centers modulating various organic functions. The vagus nerve can modulate local and systemic inflammation and possibly other neural pathways could also signal systemic inflammation to the brain [13].

Acute systemic inflammation alters behavior and produces unbalanced effects, such as delirium, in vulnerable individuals. Delirium has severe short- and long-term sequelae, but mechanisms remain unclear [25]. Systemic inflammation leads to deleterious effects on the brain parenchyma resulting in SAE. Brain lesions were demonstrated in postmortem tissue as well as in neuroimaging examination of septic patients. The main types of brain lesions in SAE are derived from ischemic processes, neuroinflammation, and metabolic stress, affecting the gray matter and the white matter in different brain regions, and are highly associated with increased mortality [13,26,27,28,29].

Microglia are the resident macrophages of the CNS and orchestrate the intracerebral inflammatory response. Microglia are the first cells to respond to any pathogenic insult in the CNS and have been associated with many neurodegenerative and brain inflammatory diseases [30,31]. Interestingly, in contrast to ischemic or traumatic brain injuries with a massive inflammatory response of macrophages and infiltrating leukocytes, SAE inflammatory activation occurs mainly in microglial cells [32]. The current hypothesis suggests a central role of microglia in neuroinflammation, particularly in the context of sepsis. This hypothesis is further discussed throughout the following sections.

## 3. The CNS Immune System

The CNS has been historically considered an immune-privileged site [33,34], mainly because the BBB—formed by tight junctions between endothelial cells, the basal lamina of these endothelial cells, and astrocytic end-feet processes—significantly reduces infiltration of macromolecules and immune cells to the parenchyma. Additionally, there are no professional antigen-presenting cells in the brain and low major histocompatibility complex (MHC) class I and II expression. Finally, the brain has long been considered to have no lymphatic drainage. This dogma, however, is being reviewed, as studies have shown that leukocytes can cross the BBB [35,36]. Furthermore, two independent studies have recently shown the presence of lymphatic vessels lining the dural sinuses located between the brain surface and the skull. Although the BBB regulates the migration of cells from the blood to the CNS, the brain is continuously monitored by resident microglia and blood-borne immune cells to detect damaging agents that would disrupt homeostasis and optimal functioning of this vital organ. The CNS has its cell populations with diverse immune functions, including microglia, astrocytes, the choroid plexus, meningeal, ventricular and periventricular macrophages, and mast cells. There are also memory T cells, a few B cells, and monocytes [37].

Astrocytes are the most abundant glial cells in the brain and play an essential role in the immune defense against pathogens. They are essential to maintain the BBB integrity, which is part of innate immunity. Following tissue injury, astrocytes become reactive and migrate to injured sites, exhibiting morphologic changes, secretion of inflammatory mediators, and form a glial scar when necessary [38]. Astrogliosis may be beneficial to neuronal survival as bioenergetics and trophic support. However, astrocytes are also sensitive to pro-inflammatory cytokines and reactive oxygen species (ROS) and are modulated by microglia. Astrocytes can assume a neurotoxic phenotype [39] and contribute to the pathophysiology of SAE since both astrocytes and microglia become activated in sepsis [40,41].

For all the above reasons, the historical view of the CNS as an immune-privileged site has been replaced by the current knowledge that the CNS can be affected by systemic infections. In the following sections, we focus on microglial cells, which are the leading neuroinflammation players.

## 4. Microglia in Homeostasis

Microglial cells were first reported in the early 1880s by Franz Nissl, a German neuropathologist [30]. At the beginning of the 20th century, Pío del Rio-Hortega proposed that microglia are a resident mesenchymal cellular component of the CNS distinct from astrocytes and oligodendrocytes [30]. Microglial cells represent between 6 and 18% of the human brain cells [42]. Like other tissue-resident macrophages, microglia are derived from the primitive hematopoiesis in the yolk sac and migrate to the brain using the circulatory system [43,44,45]. Microglia are essential to maintain brain homeostasis, exerting crucial functions in different brain development phases and the adult CNS.

Microglial cells are highly proliferative during the embryonic period and have an amoeboid morphology that facilitates migration [43]. They keep renewing themselves by local proliferation and, thereby, are almost independent of bone marrow replacement. Microglia colonize cortical proliferative zones and phagocyte neuronal progenitor cells during development, regulating the number of mature neurons [31]. The recruitment of microglia to germinal zones depends on the release of the chemokine CXCL12 by intermediate progenitors [46]. Microglia phagocyte apoptotic cells and induce neuronal cell death by releasing mediators that trigger apoptosis or phagocytosis of viable neural precursors [43]. Pharmacological lowering of the microglial cells number inversely correlates with the number of neuronal progenitors, ultimately affecting CNS development [31]. Microglia can also promote neuronal survival and neurogenesis. Microglial cells underlying the subventricular zones support neurogenesis by mechanisms that involve inflammatory cytokines such as interleukin-1β (IL-1β), IL-6, tumor necrosis factor-α (TNF-α), and interferon-γ (IFN-γ) [43,45].

Microglia are also crucial in synaptic refinement during development, eliminating inappropriate synapses by removing dendritic spines [47]. Synaptic pruning by microglia is necessary for normal brain development. In the synaptic pruning process, neuron upregulation of fractalkine (CX3CL1) signals microglia through activation of CX3CR1, and deficits in this receptor are associated with susceptibility to seizure [47]. Microglia also have a role in synaptic regulation by preventing excitotoxicity induced by glutamate [48,49]. These findings indicate a crucial role of microglia in synaptic plasticity and behavioral adaptation to the environment and chronic stress [50,51].

In a healthy adult brain, microglia present lengthy ramified processes that are motile and can expand and retract, and their branches constantly monitor the CNS, survey the whole system, and maintain homeostasis [52]. This surveillance phenotype is regulated by membrane proteins and soluble factors, such as ATP promoting microglial mobility through CX3CR1 signaling [53] and CD200, CD47, or γ-aminobutyric acid (GABA) [30] which conversely decreases their mobility. Some of these factors are released by neurons or expressed on the neurons’ surface, and the loss of these signals also indicates a loss of neuronal integrity [45]. Microglia protect the CNS against aggression (infectious, traumatic, ischemic, neurodegenerative), acting to establish tissue homeostasis through this surveillance function.

There is growing evidence of microglial heterogeneity among different parts of the brain. Microglia express neurotransmitter receptors that allow them to sense neuronal activity [52,54]. Differences in BBB leakiness, white matter, and other microenvironment factors may also contribute to microglial heterogeneity [55]. These differences allow microglia to respond in a specific way to changes in the CNS, with distinct alertness depending on the location [52]. Genome-wide analysis of microglia from different brain regions of mice showed that microglia have distinct region-dependent transcriptional identities and suggested that microglia from the cerebellum and hippocampus could be more immune-vigilant than cortex and striatum microglia [56].

## 5. Microglia in Sepsis-Associated Encephalopathy

Activated microglia exhibit rapid and profound morphological, phenotypic, bioenergetic, and functional changes in pathological conditions, broadly defined as “microglia activation” [30,57]. Microglia activation is a transient response characterized by an altered secretory profile, increased phagocytic activity, and deviation from the homeostatic phenotype.

Microglia activation is consistently observed acutely in sepsis, both in experimental models and septic patients [13,58]. Activated microglia express high levels of the 18-kDa translocator protein that can be measured in the human brain with the positron emission tomography (PET) radiotracer [11C] PBR28. LPS administration increased [11C] PBR28 binding, demonstrating microglia activation in human brains. This activation was associated with increased blood levels of inflammatory cytokines, vital sign changes, and sickness symptoms [59].

In a case-control study of 13 patients who died of sepsis, there was an increase in CD68 expression in the cortex compared to the control, and more amoeboid microglia were observed [60]. Similarly, in a postmortem case-control study of patients with delirium, there was an increase in microglial markers CD68 and HLA-DR compared to age-matched controls, suggesting that microglia activation could be related to delirium [61]. In a prospective study of 17 patients who died from septic shock, hippocampal tissue was assessed for a neuropathological study that observed increased microglia activation and apoptosis in septic patients with hyperglycemia [62].

The pathologic outcome of infections is a direct consequence of the extent of metabolic dysfunction and damage imposed on tissues that sustain host homeostasis [63]. Experimental models provide a deeper understanding of cellular response and the molecular mechanisms associated with organ dysfunction in sepsis. Oxidative stress and microglia activation have been consistently detected acutely in sepsis [12,58,64]. Genetic manipulation or pharmacological inhibition of pathways contributing to neuroinflammation and oxidative stress can protect survivors from neurocognitive impairments, indicating that neuroinflammation and oxidative stress are critical brain dysfunction mechanisms in SAE [41,65,66].

## 6. Microglia Metabolic Adaptations

Most recent studies indicate that immune cells react to their environment by reprogramming intracellular metabolic pathways that concomitantly modify immune function. This metabolic flexibility intimately connects with activation states and polarization of myeloid cells, including microglia. Microglia metabolism adapts to brain energy metabolism changes, and this metabolic reprogramming regulates microglial polarization, thereby impacting pathological inflammatory responses in the brain [67].

Brain energy metabolism and substrate availability vary under normal and disease states. Recent experimental studies have demonstrated an increase in brain glucose metabolism acutely in sepsis. A study with a porcine model of resuscitated sepsis and cerebral multimodal monitoring observed that animals developed a hyperdynamic state, increased blood lactate, and increased brain/serum glucose rates despite progressive hypoglycemia, indicating increased glucose uptake by the brain in sepsis [68]. Another study with lipopolysaccharide (LPS)-induced murine systemic inflammation model of SAE used PET and single-photon emission computed tomography (SPECT) and different radiotracers to assess brain glucose metabolism, perfusion, neuronal damage, and microglia activation. The authors observed a significant reduction in perfusion and enhanced [18F] fluorodeoxyglucose (FDG) uptake, a glucose analog. Widespread microglia activation was observed by PET and immunohistochemistry, suggesting that processes emerging during neuroinflammation may cause increased uptake of radiotracers and not neuronal hypermetabolism [69].

After inflammatory activation, immune cells’ metabolism shifts from oxidative phosphorylation to aerobic glycolysis, a phenomenon similar to the Warburg effect described initially in tumor cells [70,71]. Even in the presence of oxygen, some cells enhance glycolysis to feed other non-mitochondrial pathways that contribute to macromolecular synthesis, cell proliferation and migration, cytokine synthesis and secretion, phagocytosis, and the oxidative burst. It is now clear that ATP synthesis is not the only determinant of cell metabolism. The intermediates of glycolysis and Krebs cycle that rise when mitochondrial respiration diminishes, are required for anabolism and have been described to participate in and regulate inflammation and immune response [72,73].

Like most other cell types, microglia express gene products required for glycolytic and oxidative metabolism. Evidence suggests that microglia increase aerobic glycolysis and decrease respiration when activated by diverse inflammatory stimuli [74]. Treatment of primary microglia in vitro with 2-DG—a blocker of the glycolytic pathway—decreases TNF-α production via nuclear transcription factor κB (NF-kB) inhibition resulting in cell death [75,76] and microglia cultured with increasing glucose concentrations enhance TNF-α production [77].

Glucose is an essential fuel for microglia, and it enters the cell through different transporters (GLUTs). Microglia predominantly express GLUT3 and the fructose transporter GLUT5, but under inflammatory conditions, GLUT1 expression is upregulated to increase glucose uptake and promote glycolysis [57]. However, if there is a decrease in glucose supply, microglia can adapt to other energetic substrates, such as glutamine or fatty acids [20]. Inside the mitochondria, glutamine is converted to glutamate and is further metabolized [67].

Classical activation of microglia with LPS/IFN-γ (M1-polarized) decreases mitochondrial content and shifts metabolism from oxidative phosphorylation to glycolysis. Alternatively, activated microglia with IL-4/IL-13 (M2-polarized) resulted in enhanced mitochondrial content and function [78]. M1 microglia increase glucose uptake, the expression of proteins involved in glycolysis, and produce more lactate, and these processes are modulated by NO [74,79]. NO produced during inflammation is an inducer of metabolic shift through the reversible inhibition of the mitochondrial electron transport chain, inducing signaling cascades and gene transcription that upregulate glycolysis [80,81] (Figure 2).

Mitochondria are essential for cell viability and are involved in energy production and calcium metabolism, iron homeostasis, and cell signaling pathways of inflammation and apoptosis [82]. Mitochondrial damage leads to cytochrome C release into the cytosol, which initiates the apoptotic cascade and eventually cell death [82]. Oxidized mitochondrial DNA released in the cytosol activates the NLRP3 inflammasome [83]. Similarly, purinergic P2x7 receptors, ATP-gated ion channels expressed by virtually all immune cells, sense extracellular ATP, and its engagement results in loss of mitochondrial membrane potential and enhanced mitochondrial ROS and further contributes to inflammasome activation [84,85].

Mitochondrial dysfunctions have been described in different cell types and tissues in sepsis, including the brain [86,87]. Sepsis-induced mitochondrial dysfunction and oxidative stress compromise brain cell viability, inducing neuronal hyperexcitability, impaired neurotransmission, and eventually apoptotic cell death, which may contribute to cognitive impairments in SAE [5,13,88].

## 7. Molecular Mechanisms That Regulate Microglial Immunometabolism

Inflammation and hypoxia are intimately associated processes. Hypoxia induces inflammation, and inflamed tissue can become hypoxic. There is a combination of systemic inflammation, hypoxemia, and eventual tissue hypoxia in sepsis, inducing signals for HIF-1α accumulation. Activation of HIF-1α plays a central role in the inflammatory response as it stimulates IL-1β production, increases glycolysis, supports cell activity and survival in hypoxic-inflammatory environments [89] (Figure 2).

Metabolic reprogramming of microglia is dependent on the mTOR/HIF-1α pathway [90]. The mTOR pathway is a nutrient-sensing mechanism that regulates glucose metabolism according to the cell’s energy status. Activation of Akt contributes to mTOR’s phosphorylation required for LPS-induced glycolysis as blockage of mTOR activity suppresses the LPS-induced metabolic shift and the LPS-induced production of pro-inflammatory cytokines. Phosphorylation of mTOR upregulates the expression of HIF-1α [91].

Evidence from experimental models of neurodegenerative diseases shows that the mTOR/HIF-1α pathway is central to regulating microglial activity. Modulation of mTOR/HIF-1α pathways regulates microglia’s immune response in sepsis [92]. Microglial cells activated by LPS increased intracellular HIF-1α and resulted in metabolic reprogramming, shutting down oxidative phosphorylation and increasing aerobic glycolysis [93,94,95]. HIF-1α-deficient microglia decreased ROS production and TNF-α secretion and impaired phagocytosis [90,96] (Figure 2).

Microglia express the glucose-dependent enzyme nicotinamide adenine dinucleotide phosphate (NADPH) oxidase, or NOX2, a superoxide-generating system of pathogen killing and cell signaling. Glucose oxidation by the pentose phosphate pathway (PPP) generates NADPH, the substrate of NOX2, which transfers electrons from NADPH to oxygen, generating the superoxide anion. Glucose metabolism controls NOX activation by the nicotinamide adenine dinucleotide (NADH)-dependent transcriptional co-repressor C-terminal-binding protein (CtBP) that affects NF-κB signaling and the expression of inducible nitric oxide synthase (iNOS) [74,97]. Inhibition of NADPH oxidase promotes alternative and anti-inflammatory microglia activation during neuroinflammation [98] (Figure 2).

## 8. Microglia Immune Activation

### 8.1. Activation Phenotypes

Activated microglia assume diverse phenotypes, generally classified as “classical activation”, or M1-polarized microglia, and “alternative activation”, or M2-polarized microglia. The changes in microglial phenotypes are intimately associated with metabolic shifts of microglial cells [67,99]. Classical (M1) activation is induced by LPS or IFN-γ and leads to the activation of transcription factors NF-κΒ and STAT1 and increased expression of CD86 and MHC-II. The increase in NOX2 and iNOS produces a burst of ROS and reactive nitrogen species (RNS) and the release of pro-inflammatory cytokines, such as IL-1α, IL-1β, TNF-α, IL-6, IL-12, IL-23, chemokines CCL2 and CCL20, receptor CCR2, the macrophage receptor with collagenous structure (MARCO), and COX2. Alternative (M2) activation is induced by IL-4/IL-13 and produces a phenotype of resolution of inflammation and tissue repair associated with mitochondrial oxidative metabolism [99]. M2 microglia upon stimulation with IL-4/IL-13 initiate activation of STAT6, shifting the cells towards an anti-inflammatory phenotype with an increase in arginase (Arg-1), lectin Ym1, CD163, TREM2, and mannose receptor and release of anti-inflammatory factors IL-4, IL-10, IL-13, IL-1RA FIZZ1, and PPARγ. Exposure of microglia to anti-inflammatory IL-10 activates STAT6 and shifts the cells to a primary immunosuppressive state with an expression of CD206 and the release of IL-10, TGF-β, FIZZ1, and PPARγ [99,100].

### 8.2. Microglial Immune Receptors

The innate immune system is a coordinated cellular defense response aiming to detect and eradicate pathogens or sterile insults. Pathological insults are recognized by membrane pattern recognition receptors (PRRs), such as Toll-like receptors (TLRs) and the cytoplasmic Nod-like receptors (NLRs). Microglia represent the primary cell type in the CNS to express numerous PRRs responsible for the early recognition of PAMPs and host- or environment-derived DAMPs [101,102]. TLR2 and TLR4 have been considered the main ones involved in sepsis neuroinflammation [103,104]. Activation of TLR triggers the activation of cellular pathways responsible for the assembly of the inflammatory response, leading to the displacement of NF-κB from the cytoplasm to the nucleus inducing the transcription of genes TNF-α, IL-1, IL-6, IL-12, and IFN-γ [105]. As we discuss below, some NLR family members, when stimulated by PAMPs or DAMPs, form multicomplex cytosolic proteins critical to innate immune and inflammatory response called inflammasomes [102].

### 8.3. Inflammasome Activation

Inflammasomes are multicomplex cytosolic proteins critical for activation of the innate immune response by recognition of pathological insults. The inflammasome structure involves several subunits, including sensor NLRs, the adaptor apoptosis-associated speck-like protein containing a caspase recruitment domain (ASC), and a cysteine protease pro-caspase-1 as the effector molecule [106]. Multiple sensors detect stimuli for inflammasomes, including NACHT, LRR, and PYD domains-containing protein 1 (NLRP1), NLRP2, NLRP3, NLR family CARD domain-containing protein 4 (NLRC4), and absent in melanoma 2 (AIM2) [107,108]. Canonically, the inflammasome sensor recruits caspase-1 with the adapter protein ASC. Subsequently, caspase-1 mediates the cleavage of pro-IL-1β and pro-IL-18 into their active mature form [109]. Amongst all the inflammasomes described, NLRP3 is the most abundant inflammasome in the CNS and one of the key contributors to microglia activation and neuroinflammation in a broad spectrum of nervous system disorders [109,110] (Figure 2).

A range of physiological stress factors can induce NLRP3 inflammasome activation. These stressors include ATP [111], members of the family of NADPH oxidase enzymes such as NOX2, mitochondrial ROS [112], β-amyloid [110], α-synuclein [113] besides crystalline and aggregated substances such as asbestos, silica, and uric acid crystals [114,115]. Secreted IL-1β and IL-18 cytokines stimulate several signaling pathways, amplifying inflammation and perpetuating cell damage or inducing a cell death process called pyroptosis [116,117]. Many cell types in the brain express IL-1β receptors (IL-1βR), such as astrocytes, neurons, endothelial and ventricular cells, being potential targets of inflammasome activation and IL-1β signaling [118] (Figure 2).

Recent evidence demonstrates that NLRP3/IL-1β activation is associated with the severity of SAE and that inhibiting the adverse effects of NLRP3 inflammasomes may be a good strategy to avoid excessive inflammation during sepsis.

## 9. Crosstalk between Microglia and Other Cell Types

Interactions between activated microglia and other CNS cells contribute to neuroinflammation. Liddelow et al. [39] demonstrated that LPS-activated microglia induce a shift in the astrocyte phenotype from a neuroprotector to a neurotoxic profile (A1-polarized astrocytes). A1 astrocytes lose their ability to promote neuronal survival and tissue repair contributing to neuronal and oligodendrocyte cell death. The secretion of IL-1α, TNF-α, and C1q by activated microglia was necessary and sufficient to induce A1 astrocytes [39] (Figure 3).

IL-33 is a cytokine released in the brain by astrocytes and oligodendrocytes. Cao et al. [119] demonstrated an increase in IL-33 mRNA and protein levels produced by astrocytes after intracerebroventricular injection of LPS. IL-33 derived from astrocytes enhances microglia’s inflammatory response to LPS and mediates cerebral endothelial activation and subsequent leukocyte recruitment. IL-33 knockout mice had reduced leukocyte recruitment and neutrophil infiltration and decreased microglia activation. BV2 cells, a murine microglial cell line treated with LPS and IL-33, produced higher TNF-α and IL-6 [119].

Microglia also interact with the blood–brain barrier. In vitro BBB models treated with LPS showed increased permeability of the endothelium. This effect was reversed with treatment with an NADPH oxidase inhibitor, suggesting that ROS production by activated microglia through NADPH oxidase induces the BBB dysfunction [120,121].

Besides astrocytes and the BBB, there is essential crosstalk between microglia and neurons. The microglial fractalkine receptor CX3CR1 plays a fundamental role in the communication between microglia and neurons by binding to neuronal fractalkine, namely, CX3CL1 [122]. Accumulating evidence suggests that the CX3CL1–CX3CR1 axis is involved in homeostatic suppression of microglia activation and regulation of chemoattraction and synaptic plasticity, inhibiting microglia-mediated inflammatory responses and neurotoxicity [123]. Studies in vitro demonstrated that CX3CL1 stimulation blocks LPS-induced release of pro-inflammatory cytokines such as IL-1β, TNF-α, NO, and IL-6 by microglia [124,125]. CX3CR1 knockout mice showed intense microglia activation after intraperitoneal LPS injection and numerous annexin V-positive cells with neuronal morphology in the cortex and hippocampus [126].

Non-neural cells of immune origin such as mast cells may also release pro-inflammatory mediators that act on microglia. Neuroinflammation triggers the expression of ligands and their receptors, facilitating intercellular communication. Activated mast cells secrete a plethora of vasoactive, pro-nociceptive, and pro-inflammatory mediators (biogenic amines, cytokines, enzymes, lipid metabolites, and ATP, neuropeptides, growth factors, and NO) [127]. Zhang et al. cocultured a mast cell line (P815) with primary microglia and showed that LPS-stimulated mast cells activate microglia and subsequent production of TNF-α and IL-6 via the MAPK signaling pathway [128].

## 10. Molecular Mechanisms of Neuronal Toxicity in SAE

In neuroinflammation, activated microglia lose their original ability to maintain synapses. As microglia respond to inflammation by changing phenotypes, they reduce the frequency of contact with neurons [129]. Some authors have reported neuronal cell death associated with microglia activation in the postmortem brain of septic patients [26,27] and septic animals [40,130,131]. However, neuronal cell death is not always observed in experimental studies in sepsis [132] and is probably not associated with the neurological impairments in SAE. Activated microglia act through different mechanisms to induce neurotoxicity, regulating neurotransmission and causing synaptic dysfunction among other effects.

Evidence shows that microglia have a role in regulating mood and affective disorders, including depression, among the long-term mental health disorders found in sepsis survivors [5,133]. Recent work has demonstrated that microglia activation in the striatum is associated with increased firing of striatal medium spiny neurons and induces depressive-like behavior in mice. This effect was dependent on microglia activation after LPS injection. This inflammation-induced behavior depends on IL-6 signaling acting in an autocrine way in microglia and microglial prostaglandin signaling acting on striatal medium spiny neurons [134].

In the following sections, we discuss the main mechanisms of neuronal damage induced by activated microglia, summarized in Figure 3 and Table 1.

### 10.1. Cytokines

Dysregulation of microglial cytokine production could be harmful to defense mechanisms and disturb neural cell functions, as they are sensitive to cytokine signaling, resulting in direct neurotoxicity, as we discuss further. Higher inflammatory cytokines and chemokines are detected in the brain tissue of septic animals [66,135] and human patients [136]. The treatment of septic rats with minocycline, a known microglial inhibitor, decreases inflammatory cytokines in the brain and prevents long-term neurocognitive impairments [137]. On the other hand, depletion of microglial cell populations during severe sepsis was associated with early exacerbation of brain and systemic inflammation, and repopulation could revert this condition [138].

Recent findings suggest that activated microglia express a functional receptor for IL-23 [139] and that IL-23 may mediate inflammatory function in microglial cells by inducing IL-17 production in an autocrine manner [140]. Activated microglia produce IL-1β and IL-23, which increase the secretion of IL-17A, creating a vicious circle of sustained amplified inflammatory response. Signaling through the interaction of IL-17A and IL-17R on microglia could induce the secretion of pro-inflammatory cytokines IL-6 and MIP-2, NO, adhesion molecules, and neurotrophic factors by microglia [140]. The neutralization of peritoneal IL-17A markedly improved the prognosis of severe septic mice by decreasing neutrophil infiltration and pro-inflammatory cytokines [141]. A recent study showed that the IL-17A/IL-17R pathway blockade suppressed microglia activation and neuroinflammation, preventing cognitive impairment in experimental sepsis [135].

IL-1β concentration is significantly increased in sepsis in the peripheral and CNS [66,142]. Experimental studies revealed that the glial cells’ secretory profile determines the transitory synaptic deficit associated with cognitive impairment in sepsis survivors. Microglia activated with LPS produce a specific secretory profile distinct from that of astrocytes. LPS-stimulated microglia elicit synaptic elimination, and this effect is dependent on the IL-1β secreted by microglial cells [143]. This study provides strong evidence that IL-1β derived from activated microglia is responsible for the synaptic deficits observed in sepsis. Another study showed that IL-1β could induce synaptic loss by mechanisms that require both presynaptic and postsynaptic terminals [144]. Alternatively, IL-1β-mediated inhibition of excitatory synapses in SAE can be carried out by inserting inhibitory synaptic receptors (γ-aminobutyric acid type A receptor GABAAR) into neuronal membranes [145]. Interestingly, IL-1β produced by M1 microglia negatively interferes in the formation of synapses by inhibiting the expression of the synaptogenic cytokine IL-10 [146].

Cytokines produced by M1 microglia, such as TNF-α, can directly induce neuronal death [147]. TNF-α is a pro-inflammatory cytokine that has physiological and pathophysiological roles. Although astrocytes and neurons can produce TNF-α, microglia are the primary source of this cytokine during neuroinflammation [148]. TNF-α induces glutamate release that acts on microglial glutamate receptors to induce more TNF-α production in a vicious circle [149,150]. In parallel, TNF-α induces glutamate release by astrocytes and harms astrocyte glutamate uptake, increasing the extracellular glutamate levels, contributing to excitotoxicity. TNF-α potentiates excitotoxicity by enhancing excitatory synaptic force through increased AMPA (α-amino-3-hydroxy-5-methyl-4-isoxazolepropionic acid) and NMDA receptor expression on neurons, leading to neuronal loss [151].

### 10.2. Inflammation-Mediated Glutamate Neurotoxicity

Glutamate is the most abundant excitatory neurotransmitter in the brain regulating neuronal excitability and synaptic transmission, which constitute the main basis for information transfer in the CNS. In CNS disorders, the mechanisms that control the extracellular glutamate become dysfunctional and insufficient to buffer high levels of glutamate concentration. Glutamate accumulation in the brain may lead to neuronal excitotoxicity. Besides, activated microglia can release high amounts of glutamate, contributing to excitotoxicity during brain insults [149,150,152].

Glutamate-mediated neurotoxicity occurs through the N-methyl-D-aspartate receptor (NMDAR) activation. Excessive MNDAR activation in neurons promotes a massive influx of Ca^2+^, leading to increased production of ROS. Ca^2+^-induced ROS production decreases the mitochondrial respiratory rate, leading to impairment of ATP generation and neuronal cell death [153].

In experimental models of sepsis, the animals showed an imbalance in neurotransmitter glutamate levels [68,154]. Other studies demonstrated that glutamate release inhibition reduced the cognitive deficit and increased survival of septic animals [155].

Furthermore, microglia express almost all glutamate receptors, which can induce a neurotoxic phenotype depending on the glutamate receptor activated [54,156]. Group II of metabotropic glutamate receptors (mGluR) and ionotropic receptors, such as AMPA, kainate, and NMDA, are often associated with the neurotoxic microglial phenotype. Activation of those receptors on microglia triggers M1 polarization inducing microglial ameboid morphology [157], proliferation and migration [158], mitochondrial depolarization, caspase-3 activation, production of TNF-α, NO [159], and inflammatory cytokines, such as IL-1β and IL-17, that led to microglial and neuronal damage [160,161,162].

### 10.3. Oxidative Stress Pathways

The primary sources of cellular reactive oxidant species are the mitochondria and the NADPH oxidases, a family of enzymes responsible for superoxide generation via reduction of molecular oxygen. In the nervous system, the main isoforms are NOX1, NOX2, and NOX4. NOX2 is the phagocytic isozyme that is upregulated in phagocytes during inflammation [163].

Studies in experimental models of sepsis demonstrated that NOX2-derived ROS are essential to microglia activation and neuroinflammation in sepsis. Knockout mice for gp91, a protein essential for NOX2 activity, had impaired microglia activation and neuroinflammation after sepsis. Treatment with the NOX2 inhibitor apocynin had both antioxidant and anti-inflammatory effects in the mouse model of SAE, preventing cognitive impairments in the surviving animals [41]. Recent studies have demonstrated that NOX2 contributes to glial activation with a subsequent reduction in the expression of BDNF, synaptic dysfunction, and cognitive deficits after systemic inflammation in an LPS-injected mouse model. They provide evidence that NOX2 might be a promising pharmacological target to protect against synaptic dysregulation and cognitive impairment following systemic inflammation [164].

Nitric oxide (NO) produced by iNOS is also deleterious to neurons. In an endotoxemia model, iNOS expression increases rapidly in the brain and mainly in microglial cells [40]. Microglia express iNOS, and the genetic deficiency of iNOS protects septic animals from cognitive impairments [132]. ROS, NO, and derived reactive species molecules cause peroxidation of membrane lipids, which results in axonal damage and neurological deficits (Figure 3).

**Table 1 pharmaceuticals-14-00416-t001:** Summary of the main inflammatory molecules released by the microglia involved in the pathophysiology of sepsis-associated encephalopathy.

Inflammatory Molecule	Functions	References
Cytokines		
IL-1β	Pro-inflammatory cytokine secreted by microglia and infiltrating leukocytes. Initiates the host inflammatory response, induces synaptic dysfunction, suppresses hippocampal LTP; induces sickness behavior	[16,143,165,166]
IL-18	Pro-inflammatory cytokine; induces the release of pro-inflammatory cytokines such as IL-1β, IL-6, IFN-γ, and IL-18 by glial cells. Induces sickness behavior, loss of appetite, sleep, and inhibition of LTP	[16,167,168]
IL-6	Pleiotropic pro-inflammatory cytokine; stimulates migration of leukocytes, regulates the production of chemokines and expression of adhesion molecules, induces sickness behavior. High levels of IL-6 are strongly associated with mortality	[166,169,170]
IL-12	Evokes neuroinflammation; expressed by microglia; involved in changes of the metal status; induces production of IFN-γ from NK and activated T cells	[170,171]
IL-17	Induces the secretion of pro-inflammatory molecules (IL-1β, IL-23, IL-17, IL-6, MIP-2, NO), adhesion molecules, and neurotrophic factors by microglia; induces glial activation, microvascular pathology, and enhances neuroinflammation	[135,141,172]
IFN-γ	Upregulates cell surface molecules MHC class I and II, intercellular adhesion molecule I (ICAM-I), LPS receptor (CD14), Fc and complement receptors. Induces changes in the proteasome composition and release of cytokines (TNF-α, IL-1, and IL-6), NO, and complements (C1q, C2, C3, C4)	[104,173]
TNF-α	Pro-inflammatory cytokine; induces BBB disruption, infiltration of neutrophils, astrocytosis, and apoptosis of brain cells. Stimulates autocrine microglia activation and glutamate release by microglia and astrocytes and inhibits glutamate uptake. Suppresses hippocampal LTP	[148,165,170,174]
Chemokines		
CCL2 (MCP-1) CXCL8 (IL-8) CXCL10 (IP-10), CXCL12 (SDF-1), CCL13 (MCP-4), CCL22 (MDC) CCL3 (MIP-1α)	Chemotatic cytokines (chemokines) that induce leukocyte migration, increase BBB permeability allowing infiltration of leukocytes; chemoattractants to neutrophils and microglia; produced in several brain regions, released by activated microglia	[66,136,175,176,177]
Reactive oxidant species		
ROS, RNS, RSS	Mediators of oxidative stress; perform oxidation, nitrosylation, nitration, and sulfuration/polysulfidation reactions with endogenous molecules; change structure and function of proteins; promote lipid peroxidation altering membranes permeability, induce axonal damage and cytotoxicity; regulate gene transcription, ion transport, intermediary metabolism, and mitochondrial function; contribute to inflammasome activation	[41,64,65,178]
NO	Neurotoxic, vasodilator, mitochondrial inhibitor; gaseous signaling molecule; killing of pathogens	[32,130,132]
Neurotransmitters		
Glutamate	Secreted by activated microglia and astrocytes; induces excitatory synapses; in high concentrations induces excitotoxic neuronal cell death; induces chemotaxis of microglia	[68,153,154,155,158,179]
ATP	Secreted by activated microglia, induces microglia chemotaxis, activation, and phagocytosis	[156,180]
Prostaglandins		
PGE2	Potent inflammatory mediator; induces cytokines secretion, vasodilation, endothelial permeability and BBB disruption	[181,182]
Matrix metalloproteinases		
MMP2, MMP3, MMP8, MMP9, MMP12, MMP14	Secreted by activated microglia, degrade the extracellular matrix contributing to tissue injury. MMP8 modulates TNF-α activation and stimulates the production of IL-6 and NO. MMP-3 and MMP-9 regulate IL-1β, IL-1Ra, iNOS, and IL-6 gene expression at the transcriptional level and that of TNF-α at the post-transcriptional level.MMP-2 and MMP-9 are associated with increased BBB permeability, and inhibition of MMP-9 and MMP-2 improves acute cognitive alterations associated with sepsis.	[183,184,185]

CCL (C–C motif chemokine ligand), CXCL (C–X–C motif chemokine ligand), IP-10 (interferon gamma-induced protein 10), LTP (long-term potentiation), matrix metalloproteinases (MMP), macrophage-derived chemokine (MDC), macrophage inflammatory proteins (MIP), monocyte chemoattractant protein-1 (MCP-1), PGE2 (prostaglandin E2), reactive sulfur species (RSS), stromal cell-derived factor 1 (SDF1).

## 11. Conclusions

Sepsis-associated encephalopathy is a frequent neurological dysfunction that manifests acutely and is associated with mortality and long-term psycho-cognitive sequelae. There is no specific treatment to prevent or reverse SAE. Current evidence points to a central role of neuroinflammation in the pathophysiology of SAE, and microglial cells are major players. Understanding the molecular mechanisms of microglia activation and neurotoxicity may provide new therapeutic targets and opportunities for SAE treatment.

## Figures and Tables

**Figure 1 pharmaceuticals-14-00416-f001:**
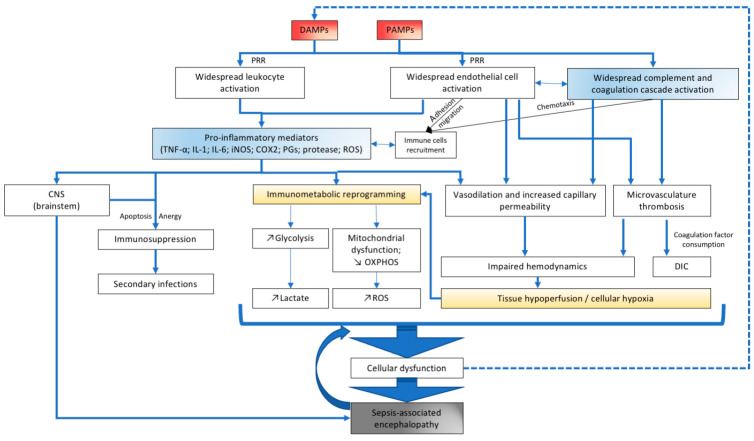
Overview of the pathophysiology of sepsis-associated encephalopathy. DAMPs (damage-associated molecular patterns), PAMPs (pathogen-associated molecular patterns), PRR (pattern recognition receptors), TNF (tumor necrosis factor), IL-1, IL-6 (interleukins 1 and 6), iNOS (inducible nitric oxide synthase), COX2 (cyclooxygenase 2), PGs (prostaglandins), ROS (reactive oxygen species), CNS (central nervous system), OXPHOS (oxidative phosphorylation), DIC (disseminated intravascular coagulation).

**Figure 2 pharmaceuticals-14-00416-f002:**
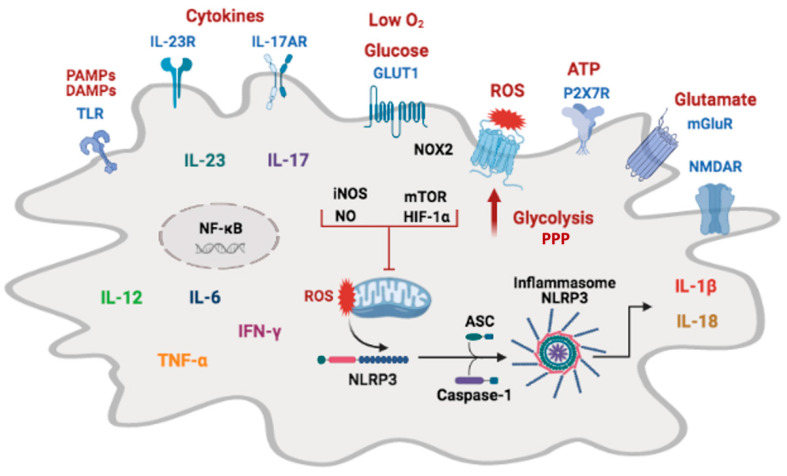
Molecular mechanisms of microglia activation in sepsis-associated encephalopathy. Classically (M1) activated microglia express Toll-like receptors (TLR) that recognize pathogen-associated molecular patterns (PAMPs) and damage-associated molecular patterns (DAMPs) and trigger NF-κB-dependent pro-inflammatory gene expression, upregulating inflammatory cytokines such as IL-6, IL-12, TNF-α, and IL-23 and components of the inflammasome pathway. IL-23 mediates inflammatory response by inducing IL-17 production and the secretion of pro-inflammatory cytokines. Inducible NOS is upregulated in M1 microglia, and NO induces a metabolic shift to glycolysis through mitochondrial inhibition. Inflammation and hypoxia activate the mTOR/ HIF-1α pathway, inhibit mitochondrial oxidative phosphorylation, and increase glycolysis. GLUT1 is upregulated to increase glucose uptake by M1 microglia. Glucose oxidation through the pentose phosphate pathway (PPP) generates NADPH, which is the substrate for NOX2 and iNOS to produce ROS and NO, respectively. P2x7, the ATP receptor activation, and mitochondrial ROS trigger NLRP3 inflammasome activation and IL-1β and IL-18 release. Activation of microglial metabotropic (mGluR) and N-methyl-D-aspartate (NMDA) glutamate receptors (NMDAR) trigger M1 polarization. ASC, adaptor molecule apoptosis-associated speck-like protein containing a CARD.

**Figure 3 pharmaceuticals-14-00416-f003:**
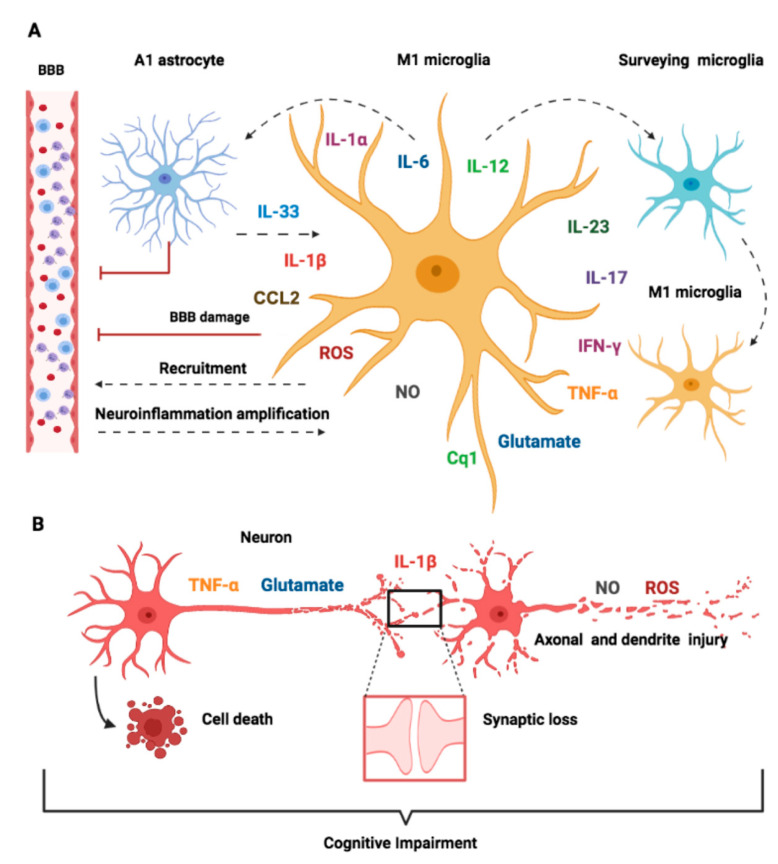
Schematic representation of molecular mechanisms of microglia-mediated neurotoxicity. Classically (M1) activated microglia induce neurotoxicity by direct and indirect mechanisms, including the induction of neurotoxic astrocytes (A1), damage of the brain endothelium, promotion and intensification of inflammation, synaptic dysfunction, neuronal injury, and cell death. All these mechanisms contribute to cognitive impairments and acute neurological dysfunctions in SAE. (**A**) M1 microglia produce pro-inflammatory molecules such as IL-1β, IL-1α, IL-6, IL-12, IL-23, IL-17, IFN-γ, TNF-α, glutamate, C1q, NO, and ROS. The secretion of IL-1α, TNF-α, and C1q by M1 microglia induces A1-polarized astrocytes and contributes to BBB disruption and IL-33 release, enhancing the inflammatory response of activated microglia. The release of ROS, NO, and CCL2 increases the brain endothelium’s permeability and mediates BBB disruption. The generation of IL-1β, IL-12, IL-23, IL-17, IFN-γ, TNF-α, and glutamate can, in turn, activate surveying microglia and recruit immune cells from the periphery to the CNS, which amplifies the inflammatory signal, creating a vicious circle of sustained and amplified neuroinflammation. (**B**) M1 microglia effects on neuronal functions. The release of TNF-α and high levels of glutamate induce neuronal excitotoxicity and cell death. IL-1β release by activated microglia causes synaptic loss. ROS and NO promote peroxidation of membrane lipids and axonal damage. BBB, blood–brain barrier; IL, interleukin; C1q, complement component 1; NO, nitric oxide; ROS, reactive oxygen species; TNF, tumor necrosis factor; IFN, interferon.

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
