# Peer review of "Neuroinflammation in Sepsis: Molecular Pathways of Microglia Activation"

_pharmaceuticals, 2021, doi:10.3390/ph14050416_

Round 1

Reviewer 1 Report

This review presents the molecular pathway of microglia activation in the neuroinflammation associated with sepsis.  Microglia is a major player of neuroinflammation in the pathophysiology of sepsis-associated encephalopathy.  This review discusses how the brain is affected by peripheral immune activation in sepsis and the role of microglia in these processes.  The review provides a helpful contribution to understanding molecular mechanisms of microglia activation in sepsis-associated encephalopathy. There are some minor points that require authors’ attention.

  1. P2 L64  Please change “SEA" for " SAE ".
  1. P14 L520  Please change “IL1b" for " IL-1b ".
  1. P14 L534  Please change “TNFa" for " TNF-a ".
  1. P15 L569  Please change “Nox-2" for " Nox2 ".
  1. P15 L571, L572  Please change “NOX-2" for " NOX2 ".

Author Response

We thank reviewer 1 for the comments. We made all the corrections suggested (marked in red in the text), and we checked the whole manuscript for spell and grammar mistakes. 

Reviewer 2 Report

My suggestions

  1. I would explain in a short paragraph (in the introduction) on the association of microglia/inflammation and other neurodegenerative diseases.
  2. Were any genetic factors identified between inflammatory molecules and SAE?
  3. A table, which summarizes the inflammatory molecules, involved in sepsis, would be nice.
  4. For Chapter 10 (Molecular mechanisms of neuronal toxicity in SAE  ), I would add an additional figure, which summarizes the toxicity mechanisms

Author Response

We really appreciate the suggestions, they certainly made great improvements to our manuscript.

We answered each question bellow:

  • I would explain in a short paragraph (in the introduction) on the association of microglia/inflammation and other neurodegenerative diseases.

We included one paragraph in the introduction explaining the association of microglia/neuroinflammation and other neurodegenerative diseases

  • Were any genetic factors identified between inflammatory molecules and SAE?

As far as our knowledge there is not strong association described for susceptibility or resistance to sepsis-associated encephalopathy.

  • A table, which summarizes the inflammatory molecules, involved in sepsis, would be nice.

We included a table at the end of the manuscript summarizing the main inflammatory mediators involved in the pathophysiology of SAE and their functions.

  • For Chapter 10 (Molecular mechanisms of neuronal toxicity in SAE), I would add an additional figure, which summarizes the toxicity mechanisms

We actually had this figure, but it was misplaced in the manuscript. We modified Figure 3 and we moved it to the end of the manuscript. Figure 3 clearly summarizes the mechanisms of neurotoxicity.

Reviewer 3 Report

The paper reviews the literature on the role of neuroinflammation in sepsis, focusing on the molecular pathways underlying microglial activation.

The paper is well written and easy to read. The different chapters are also well organised and progressively describe how the microglial system is organised, how it functions, what its physiological roles are to eventually explain its involvement in sepsis-related neuroinflammation. 

I do not find any major faults.

I finally list a few minor spelling/mistakes for the authors to address.

Figure 1: correct IL6 to IL-6 to uniform with the text

line 131: reduces

line 181: remove "and" and put ", ultimately"

line 199: decreases

line 306: format character

lines 348-349: consider revising the sentence

line 362: replace "leading" to "leads to the"

line 363: produces

line 373: shifts

line 426-427: consider rephrasing

line 432: reference for Cao et al.

line 452: uniform character CX3CR1

lines 459-460: missing ")"

line 460: Zhang reference

line 468: contributes

line 469: recrutiment of

line 471: induces

line 482: doesn't does not

line 483: act

line 508: suggest that

line 509: that may

line 589: sepsis-associated

line 589: manifests

Best regards

Author Response

We thank reviewer 3 for the comments and suggestions. We made all the changes suggested in the text (shown in red), and we also checked the whole manuscript for spell and grammar mistakes. Best regards.

Reviewer 4 Report

The manuscript by Moraes et. al, sum up recent progress in neuroinflammation and sepsis, an emerging research investigation field.

The literature cited is of important relevance and the authors are aiming to introduce their own style, interpretation as well as outlook to this interesting topic. They arrange a review that explore extensively the molecular and cellular basis of the potential sepsis mechanisms involved in the immune response of central nervous system. Moreover, they reported an update about the microglial cells as active players in sepsis-associated encephalopathy (SAE).

This is a well written review and the length of the paper is commensurate with the message.

The review may be accepted for publication in this journal.

Author Response

We thank very much the comments of the reviewer 4. 

Round 2

Reviewer 2 Report

Manuscript is acceptable now

Author Response

We thank the reviewer very much for taking the time to review our manuscript so carefully. We made a thorough revision of the text and corrected all the spell and grammar mistakes of the manuscript. Best regards.